# Analyzing Trends in Saharan Dust Concentration and Its Relation to *Sargassum* Blooms in the Eastern Caribbean

José J. Hernández Ayala [1,*] and Rafael Méndez-Tejeda [2]

1    Graduate School of Planning, University of Puerto Rico Rio Piedras Campus, San Juan 00921, Puerto Rico
2    Atmospheric Sciences Laboratory, University of Puerto Rico at Carolina, Carolina 00984, Puerto Rico; rafael.mendez@upr.edu
\*    Correspondence: jose.hernandez180@upr.edu

**Abstract:** This study investigates the temporal trends and correlations between Saharan dust mass concentration densities (DMCD) and *Sargassum* concentrations (SCT) in the tropical North Atlantic. Average DMCD data for June, July, and August from 1980 to 2022, alongside SCT data for the same months from 2012 to 2022, were analyzed using Mann–Kendall tests for trends and lagged regression models to assess whether higher Saharan dust levels correlate with *Sargassum* outbreaks in the region. A comprehensive analysis reveals a significant upward trend in Saharan dust quantities over the study period, with the summer months of June, July, and August exhibiting consistent increases. Notably, 2018 and 2020 recorded the highest mean DMCD levels, with June showing the most significant increasing trend, peaking in 2019. These findings are consistent with previous studies indicating a continuous elevation in Saharan dust concentrations in the tropical atmosphere of the North Atlantic. Simultaneously, *Sargassum* concentrations also show a notable increasing trend, particularly in 2018, which experienced both peak SCT and elevated DMCD levels. Mann–Kendall tests confirm statistically significant upward trends in both Saharan dust and *Sargassum* concentrations. Simple linear regression and lagged regression analyses reveal positive correlations between DMCD and SCT, highlighting a temporal component with stronger associations observed in July and the overall June–July–August (JJA) period. These results underscore the potential contribution of elevated Saharan dust concentrations to the recent surge in *Sargassum* outbreaks in the tropical North Atlantic. Furthermore, the results from forward stepwise regression (FSR) models indicate that DMCD and chlorophyll (CHLO) are the most critical predictors of SCT for the summer months, while sea surface temperature (SST) was not a significant predictor. These findings emphasize the importance of monitoring Saharan dust and chlorophyll trends in the Eastern Caribbean, as both factors are essential for improving *Sargassum* modeling and prediction in the region. This study provides valuable insights into the climatic factors influencing marine ecosystems and highlights the need for integrated environmental monitoring to manage the impacts on coastal economies.

**Keywords:** Saharan dust; *Sargassum*; sea surface temperatures; chlorophyll; Caribbean

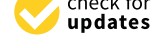



## 1. Introduction

Saharan dust, a phenomenon characterized by the transport of fine mineral particles from the arid regions of the Sahara Desert in North Africa across vast distances, exerts substantial influences on ecosystems and human activities in the tropical North Atlantic and the greater Caribbean region [1]. This natural phenomenon, also known as the Saharan Air Layer (SAL), has gained increasing attention due to its far-reaching impacts on air quality, weather patterns, and various sectors such as agriculture, health, and tourism [2–6]. The Saharan dust episodes are often associated with elevated concentrations of particulate matter (Figure 1) and the presence of various pollutants, which can lead to respiratory issues and reduced visibility [7]. Additionally, these dust particles can interact with cloud formation and solar radiation, potentially influencing local weather and climate [8].

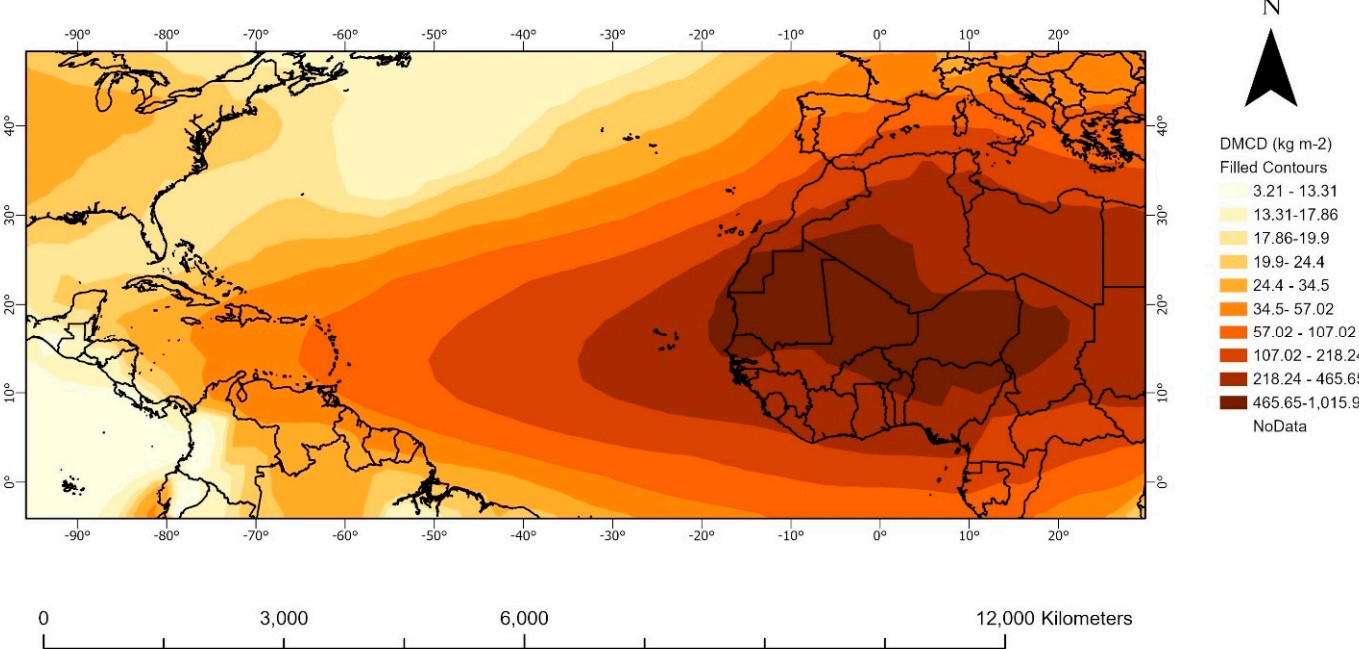

**Figure 1.** Mean Saharan dust column mass density for 28 June 2018.

As global temperatures rise due to climate change, the Sahara Desert experiences altered wind patterns and increased aridity, intensifying dust emissions [9]. This higher dust load can affect the radiative balance of the Earth by absorbing or reflecting sunlight, potentially contributing to regional and even global climate shifts [10,11]. Furthermore, the nutrient content within the transported dust can fertilize marine ecosystems, including the pelagic *Sargassum* in the tropical Atlantic [1,12,13]. While the exact mechanisms are still being studied, it is suggested that iron and other dust minerals might enhance *Sargassum* growth.

Saharan dust can have positive effects on *Sargassum*. The deposition of Saharan dust on the ocean surface can provide nutrients, including iron and phosphorus, which are essential for the growth of *Sargassum* [14,15]. These dust-derived nutrients act as fertilizers, stimulating primary productivity and potentially enhancing *Sargassum* growth rates. Iron, in particular, is a micronutrient that can be limiting in some oceanic regions, and the input of iron-rich Saharan dust can alleviate this limitation for *Sargassum* and other marine organisms [16].

In recent years, unprecedented SAL events have coincided with *Sargassum* blooms in the tropical waters of the North Atlantic, but little is known about whether Saharan dust deposition contributes to higher concentrations of the algae. Research on the relationship between SAL and pelagic *Sargassum* in the tropical North Atlantic has shed light on their interactions and potential influences, with some studies showing that these dust particles can affect the growth and distribution of the algae (12), even triggering the appearance of large-scale blooms in the Caribbean Sea [17,18].

Some studies suggest that atmospheric dust, which contains essential nutrients such as iron, phosphorus, and nitrogen, can enhance the growth of marine macroalgae, including *Sargassum* [19]. Dust-borne phosphorus also plays a crucial role in marine productivity [20]. There is empirical evidence supporting the hypothesis that nutrient-rich dust can stimulate algal blooms, identifying a correlation between dust transport events and increased *Sargassum* growth in the Caribbean Sea [21]. Such findings suggest that the deposition of nutrient-rich dust may be a significant factor in the proliferation of *Sargassum*.

Other factors that are important drivers of *Sargassum* blooms are sea surface temperature and high nutrient concentrations. Higher sea surface temperatures can promote the proliferation of *Sargassum* as warmer waters provide optimal conditions for its growth

and reproduction [22]. On the other hand, higher chlorophyll concentrations in the water column can indicate regions of enhanced primary productivity and nutrients, which may be associated with favorable conditions for the growth and proliferation of *Sargassum* populations [23]. In combination with Saharan dust, sea surface temperatures and elevated nutrient concentrations could explain why some periods reflect higher *Sargassum* blooms.

For that reason, the objective of this study is to further examine the relation between SAL events and *Sargassum* blooms to determine if higher dust deposition leads to larger algae growth in the area. This study examines temporal trends in Saharan dust events in the Eastern Caribbean to determine if high-concentration SAL occurrences have increased over time and if there is any connection with anomalous *Sargassum* blooms in the region. After examining trends in SAL events in the tropical waters of the North Atlantic, the focus shifts to examining whether those events coincide with *Sargassum* outbreaks in the region.

## 2. Data & Methods

Average dust mass column density (DMCD) data—which refers to the total mass per unit area of a specific substance, such as aerosols or gasses, extending vertically from the Earth's surface to the top of the atmosphere—were obtained from MERRA-2 (Modern-Era Retrospective Analysis for Research and Applications, version 2), a state-of-the-art atmospheric reanalysis model developed by NASA. This variable is calculated by integrating the atmospheric concentration of the aerosols over the entire vertical column above a given point on the Earth's surface. The integration involves accounting for factors such as the vertical distribution of the dust and aerosols, the density of the air, and the altitude variations within the column. The monthly average MCD data were extracted for the Eastern Caribbean region of the tropical North Atlantic (−80.5, 10, −60, 23.5) for the entire period 1980–2022 (Figure 2A). The analysis was limited to that section of the study area to compare DMCD concentrations with *Sargassum* coverage areas (Figure 2B). The DMCD data have been used by previous researchers examining trends and patterns in Saharan dust concentrations [24–26].

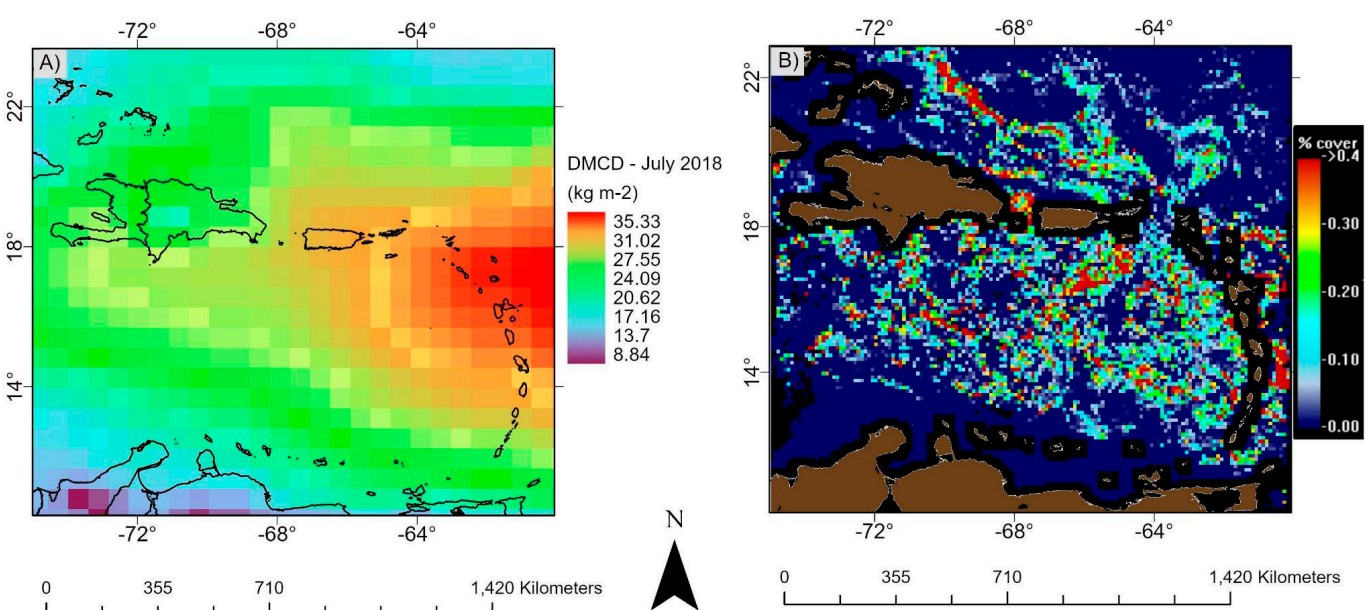

**Figure 2.** Mean dust mass column density (DMCD) for July of 2018 (**A**) and floating algae density, in terms of percentage of area covered for July 18, 2018 (**B**).

Monthly *Sargassum* concentrations (SCT) in tons for a section of the Eastern Caribbean (−80.5, 10, −60, 23.5) were obtained from the Optical Oceanography Lab at the University of South Florida [27]. The monthly average *Sargassum* in tons was extracted for June, July, and August (JJA), which have been documented to be the period with higher concentrations

of the algae [1,12]. The temporal resolution of the *Sargassum* data is from 2010 through 2022. Mann–Kendall (MK) tests for trends were performed on the DMCD and SCT to determine if the dust and *Sargassum* concentrations during the JJA period or individual months were increasing over time. The MK test is a widely used statistical test in the atmospheric sciences that allows for robust trend detection, and several studies have implemented the method to analyze trends in dust aerosols [28–30].

Mean sea surface temperature (SST) and chlorophyll (CHLO) data were also extracted for the study area to determine if those factors, in combination with DMCD, could also explain SCT variability. Mean SST data for June, July, and August were obtained from the NOAA Extended Reconstruction SSTs Version 5. Average chlorophyll concentrations for the same months and region were extracted from the Moderate-Resolution Imaging Spectroradiometer (MODIS).

The DMCD and SCT data were analyzed using simple linear regression models to determine if higher dust amounts coincided with higher *Sargassum* concentrations in the study area. The regression models between Saharan dust and *Sargassum* concentrations were performed on the individual months and the JJA period. The simple linear regression models were also performed with a lag, this was done to consider the possibility that previous Saharan dust deposition in the ocean would later enhance *Sargassum* growth in a later period. This type of regression is particularly useful when you suspect that the current value of a variable depends on its past values; for example, in this case, we hypothesize that higher Saharan dust deposition leads to higher *Sargassum* concentrations.

A one-month lag between dust arrival and *Sargassum* concentration is justified as nutrients from dust, like iron and phosphorus, take several weeks to be absorbed and utilized. This aligns with observations that phytoplankton blooms occur weeks after nutrient deposition. *Sargassum* mats can persist in regions, allowing time to utilize these nutrients [18,31].

Forward stepwise regression (FSR) procedures were used to examine how important DMCD was to SCT variability when compared to other factors like SST and CHLO. FSR is a statistical method that sequentially adds predictor variables to a regression model based on their contribution to improving the model's fit, aiming to identify the most relevant factors for explaining the variability in the response variable. In this case, SCT is the response variable that we are interested in predicting using the predictors DMCD, SST, and CHLO.

## 3. Results and Discussion

### 3.1. Trends in Saharan Dust and Sargassum Concentrations

When trends in monthly and seasonal average DMCD concentrations are examined, we find that Saharan layer dust quantities in the atmosphere of the tropical North Atlantic have been increasing over time (Figure 3A). When the summer months of JJA are analyzed together, the results show a steadily increasing trend of mean DMCD for the tropical North Atlantic for the 1980–2022 period (Figure 3A). The years with the highest mean DMCD concentrations were 2018 and 2020. Of the three summer months (JJA) with higher mean DMCD concentrations, June was found to have the most significant increasing trend in Saharan dust from 1980 through 2022 (Figure 3A). For the June series, 2019 was found to be the year with the highest mean DMCD. The individual months of July and August show increasing trends in average DMCD concentrations (Figure 3A), yet those were not as significant as the positive trend in Saharan dust quantities that June reflected. Of all of the years analyzed, July 2018 was found to be the period with the highest average DMCD. These results confirm what some studies have been suggesting, that Saharan layer dust concentrations have been increasing over time in the tropical atmosphere of the North Atlantic [9].

The time series plots for *Sargassum* concentrations also suggest that the algae have been experiencing an increasing trend in total quantity and area in the region (Figure 3B). The results for the JJA period show that *Sargassum* concentrations have been increasing from 2010 through 2022, with 2018 being the year with the highest agglomerations detected

(Figure 3B). It is important to note that 2018 was also the year with the highest DMCD concentrations in the region, so both the highest *Sargassum* agglomerations and Saharan dust quantities coincided during that period. When the individual months are examined, we find that all of them exhibited significant increasing trends in *Sargassum* concentration (Figure 3B). June and July of 2018 and August of 2021 were the periods with the highest mean concentrations of *Sargassum* in the region (Figure 3B). These results coincide with the findings of other studies that have detected a recent increase in *Sargassum* concentrations and total area covered [18], which might be associated with higher dust deposition from the Sahara [12].

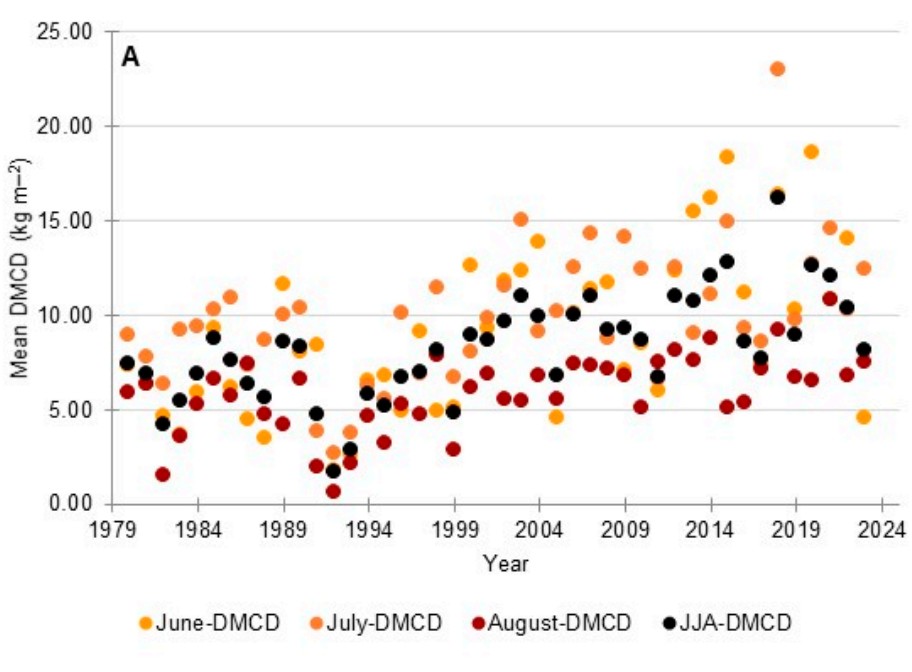

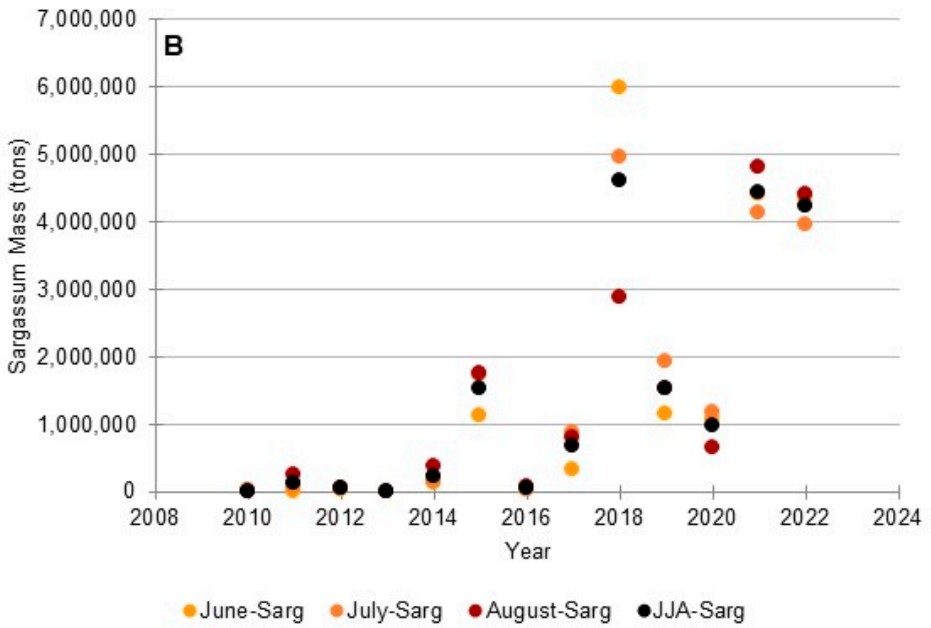

**Figure 3.** Average monthly dust mass concentrations in kg m$^{-2}$ for the 1980–2022 period (**A**) and average June, July, and August *Sargassum* concentrations in tons from 2010–2022 (**B**).

When both DMCD and SCT time series are analyzed using Mann–Kendall tests for trends, we find that Saharan dust and *Sargassum* concentrations have been increasing over time in the tropical atmosphere of the North Atlantic (Table 1). The month of June was found to have the most statistically significant increase in DMCD and SCT concentrations. When the three months are examined together, the results show that both Saharan dust and *Sargassum* concentrations reflect statistically significant trends. Even though July and August show weaker increasing trends in mean DMCD when compared to June and the JJA period, they still exhibit statistically significant trends (Table 1). The Mann–Kendall results for SCT show that all of the individual months and the JJA period exhibited statistically significant increasing trends in *Sargassum* concentrations, with June showing the strongest trend.

**Table 1.** Results of Mann–Kendall tests for trends for Saharan dust (DMCD) and *Sargassum* concentrations in tons (SCT) for different periods.

| Period (1980–2022) | Tau-Coeff Dust | *p*-Value Dust |
|---|---|---|
| June | 0.451 | 0.001 |
| July | 0.258 | 0.015 |
| August | 0.287 | 0.006 |
| JJA | 0.433 | 0.001 |
| **Period (2012–2022)** | **Tau-Coeff Sarg** | ***p*-Value Sarg** |
| June | 0.636 | 0.001 |
| July | 0.576 | 0.011 |
| August | 0.58 | 0.011 |
| JJA | 0.57 | 0.011 |

### 3.2. Statistical Modeling Results

The results of the simple linear regression models between DMCD and SCT show an overall positive correlation between the two variables (Figure 4); as Saharan dust concentrations increase, so do *Sargassum* concentrations. When the analysis is done for the JJA (Figure 4A), the results show a very strong positive correlation (R2 0.429) between DMCD and SCT. The results for the individual months show the month of July (Figure 4C) with the strongest correlation, while weaker correlations were found for June and August (Figure 4B,D). These findings suggest that the relationship between Saharan dust and *Sargassum* concentrations has a particular temporal component, with weaker correlations in June and August, and stronger correlations in July and the JJA period overall.

The results of the lagged regression models also show positive correlations between DMCD and SCT (Figure 4E,D). When the lagged regression was done using DMCD for June and SCT for July, we found a positive yet weak correlation between Saharan dust and *Sargassum* (Figure 4E). However, when the analysis was performed using DMCD for July and SCT for August, the results yielded a stronger positive correlation between the two factors (Figure 4D). These findings suggest that higher dust concentrations in June do not necessarily materialize into higher *Sargassum* quantities in July. Yet, greater amounts of dust in July show a stronger statistical relationship with elevated *Sargassum* concentrations. These results are similar to those found in previous studies [12,14], showing that higher Saharan dust concentrations likely enhanced some of the recent *Sargassum* outbreaks in the region.

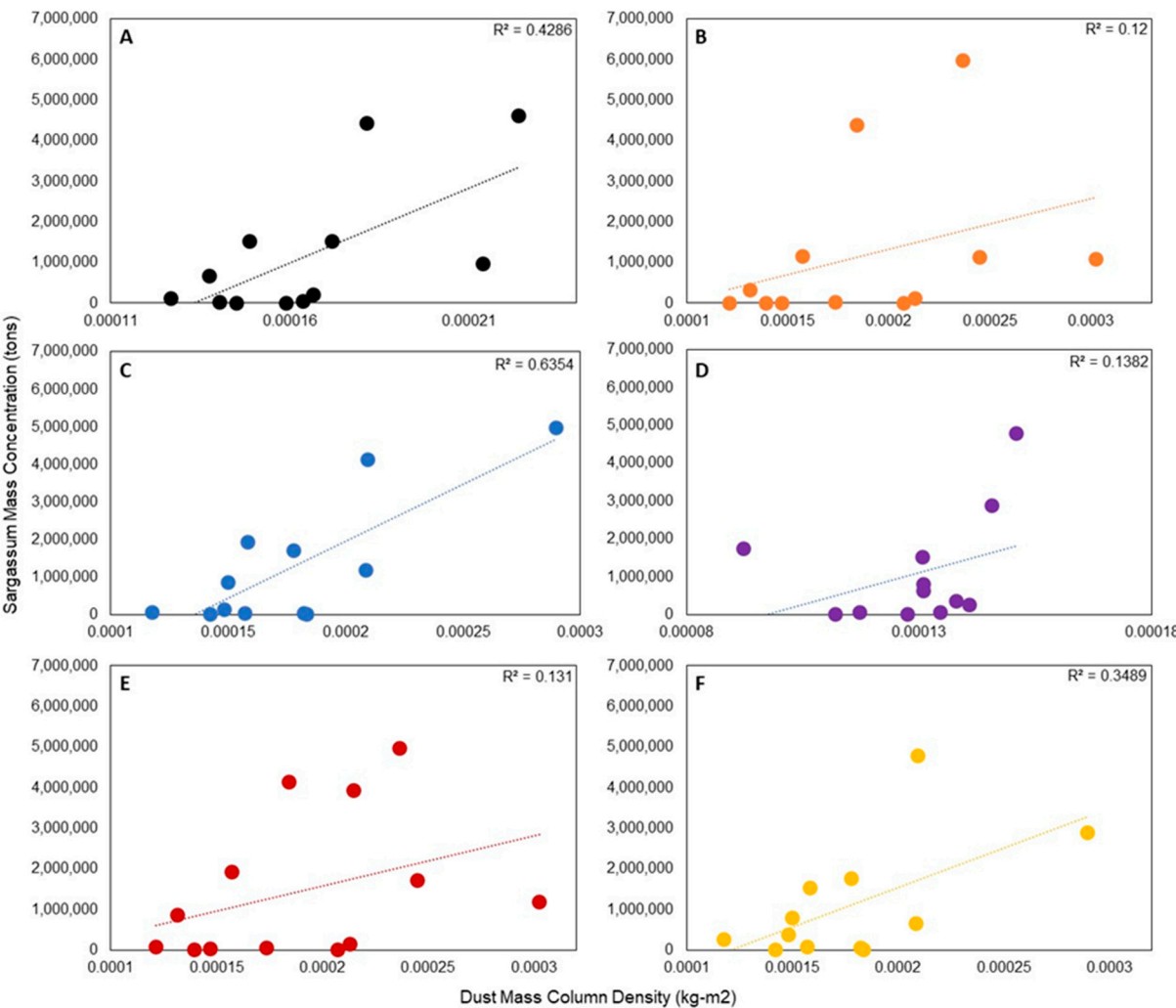

**Figure 4.** Simple linear regression models between mean monthly DMCD and total SCT data for the JJA period (**A**), June (**B**), July (**C**), and August (**D**). Lagged regression models between total *Sargassum* concentrations in tons (y-variable) and mean Saharan dust concentrations for June (dust) and July (*Sargassum*) (**E**) and July (dust) and August (*Sargassum*) (**F**).

The results of the forward stepwise regression (FSR) models suggest that DMCD is one of the most important predictors of SCT in the Eastern Caribbean region that was analyzed in this study (Table 2). The FSR results for June selected average CHLO in May as the most important factor behind higher mean SCT, suggesting that elevated chlorophyll concentrations often indicate increased phytoplankton growth. While phytoplankton and Sargassum are both primary producers and compete for nutrients, the presence of phytoplankton, indicated by elevated chlorophyll levels, suggests nutrient-rich conditions that can support Sargassum growth [18]. FSR results for July and August show DMCD as the most important predictor of SCT, suggesting that higher Saharan dust deposition during those months is the key driver of higher *Sargassum* concentrations (Table 2). The FSR results for the JJA period identified both CHLO and DMCD as the most important factors that were examined, explaining higher SCT in the region. CHLO alone accounted for a third of the variability in SCT, while the addition of DMCD into the model increased the adjusted R-square to 0.563. These results suggest that Saharan dust and chlorophyll are the most important predictors of SCT in the Eastern Caribbean region. It is important to note that SST was not selected as an important factor in the FSR models, suggesting that ocean temperatures in this region might not be as critical to higher *Sargassum* concentrations as Saharan dust and chlorophyll.

**Table 2.** Forward stepwise regression (FSR) models of mean *Sargassum* concentration and several factors, including the means of dust column mass density, sea surface temperatures, and chlorophyll.

| | **Forward Stepwise Regression Models** | | | | | |
|---|---|---|---|---|---|---|
| | *Avg. Sargassum Concentra-Tions—June* | | | | | |
| Step | Entered | Adj. R-Square | R-Square | C(p) | AIC | RMSE |
| 1 | Avg. CHLO (May) | 0.3968 | 0.3420 | 4.1159 | 413.3 | 1,678,885.8 |
| | *Avg. Sargassum Concentrations—July* | | | | | |
| Step | Entered | Adj. R-Square | R-Square | C(p) | AIC | RMSE |
| 1 | Avg. DMCD (July) | 0.4385 | 0.3875 | 1.2860 | 408.6 | 1,397,085.3 |
| | *Avg. Sargassum Concentrations—August* | | | | | |
| Step | Entered | Adj. R-Square | R-Square | C(p) | AIC | RMSE |
| 1 | Avg. DMCD (Aug) | 0.2148 | 0.1434 | 0.8582 | 411.2 | 1,548,442.6 |
| | *Avg. Sargassum Concentrations—Jja* | | | | | |
| Step | Entered | Adj. R-Square | R-Square | C(p) | AIC | RMSE |
| 1 | Avg. CHLO (JJA) | 0.3536 | 0.2948 | 7.2811 | 410.7 | 1,514,045.2 |
| 2 | Avg. DMCD (JJA) | 0.6364 | 0.5637 | 2.1574 | 405.2 | 1,190,910.5 |

## 4. Conclusions

In this study, the average Saharan dust mass concentration densities (DMCD) and mean *Sargassum* concentrations (SCT) in the tropical North Atlantic were analyzed to identify any coinciding increasing trends over the examined period. Mann–Kendall tests for trends and lagged regression models were used to examine the relationship between DMCD and SCT to determine if higher Saharan dust concentrations were associated with *Sargassum* outbreaks in the region. The analysis aimed to understand the potential impact of Saharan dust on the proliferation of *Sargassum*.

The analysis revealed a significant upward trend in Saharan dust quantities from 1980 to 2022, particularly during the summer months of June, July, and August (JJA). Notably, the years 2018 and 2020 recorded the highest levels of DMCD, with June exhibiting the most significant increasing trend, peaking in 2019. These findings are consistent with existing studies that indicate a continuous rise in Saharan dust concentrations in the tropical atmosphere of the North Atlantic, suggesting an ongoing elevation of dust levels in the region.

Simultaneously, *Sargassum* concentrations showed a parallel increasing trend, with 2018 marked by both peak SCT agglomerations and heightened DMCD concentrations. The Mann–Kendall tests confirmed statistically significant increasing trends in both Saharan dust and *Sargassum* concentrations. Additionally, simple linear regression models and lagged regression analyses established positive correlations between DMCD and SCT, revealing a temporal component with stronger correlations observed in July and the overall JJA period. These results support the hypothesis that Saharan dust contributes to Sargassum outbreaks in the tropical North Atlantic.

While this study provides valuable insights into the relationship between Saharan dust and *Sargassum* blooms, several limitations should be considered. The use of monthly average data may obscure short-term variations and specific events that could offer a more nuanced understanding of the relationship between dust deposition and *Sargassum* growth. The study's focus on a specific section of the tropical North Atlantic may not capture the full spatial variability of both dust deposition and Sargassum blooms across the entire region.

Moreover, while correlations between DMCD and SCT were identified, the specific nutrient dynamics and biological mechanisms by which Saharan dust influences *Sargassum* growth remain complex and are not fully explored. Other factors, such as ocean currents,

local nutrient sources, and anthropogenic impacts, were not comprehensively analyzed and could also play significant roles.

Future research should address these limitations by incorporating higher temporal resolution data, expanding the spatial scope of analysis, and investigating the detailed nutrient dynamics and biological mechanisms involved. Additionally, exploring the interplay between Saharan dust, local nutrient sources, and other environmental factors will provide a more comprehensive understanding of the drivers behind *Sargassum* blooms.

**Author Contributions:** Conceptualization, J.J.H.A. and R.M.-T.; methodology, J.J.H.A.; software, J.J.H.A.; validation, J.J.H.A. and R.M.-T.; formal analysis, J.J.H.A.; investigation, J.J.H.A. and R.M.-T.; resources, R.M.-T.; data curation, J.J.H.A.; writing—original draft preparation, J.J.H.A. and R.M.-T.; writing—review and editing, J.J.H.A. and R.M.-T.; visualization, J.J.H.A.; supervision, R.M.-T.; project administration, R.M.-T.; funding acquisition, R.M.-T. All authors have read and agreed to the published version of the manuscript.

**Funding:** This research received no external funding.

**Institutional Review Board Statement:** Not applicable.

**Data Availability Statement:** Average monthly dust mass column density data was obtained from the MERRA-2 (Modern-Era Retrospective Analysis for Research and Applications, version 2) https://gmao.gsfc.nasa.gov/reanalysis/MERRA-2/data_access/, (accessed on 10 August 2024). Monthly *Sargassum* concentrations (SCT) in tons for a section of the Eastern Caribbean ($-80.5$, 10, $-60$, 23.5) were obtained from the Optical Oceanography Lab at the University of South Florida (https://optics.marine.usf.edu/projects/saws.html, accessed on 30 August 2024). Mean monthly SST data were obtained from the NOAA Extended Reconstruction SSTs Version 5 (https://psl.noaa.gov/data/gridded/data.noaa.ersst.v5.html, accessed on 20 October 2023) Average chlorophyll concentrations for the same months and region were extracted from the Moderate-Resolution Imaging Spectroradiometer (https://oceancolor.gsfc.nasa.gov/resources/atbd/chlor_a/, accessed on 30 August 2024).

**Conflicts of Interest:** The authors declare no conflict of interest.

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
