# Peer review of "Analyzing Trends in Saharan Dust Concentration and Its Relation to Sargassum Blooms in the Eastern Caribbean"

_2673-1924, doi:10.3390/oceans5030036_

Round 1
Reviewer 1 Report
Comments and Suggestions for Authors
This is an interesting exercise and provides some evidence that atmospheric dust could contribute to the large Sargassum blooms in the Tropical Atlantic. However, greater justification is needed for the statistical treatment and the choice of lag in particular. Additionally, the limitations of the study are not effectively addressed. What are the biases in the data?
Consider adding better justification for the hypothesis. Which components of the dust could be increasing Sargassum growth? Is there evidence in the literature of enhanced macroalgal growth with some of the dust constituents?
Figure 3 caption units do not match units on y-axis for panel A. Also, be clear that the average Sargassum concentration plotted in panel B is the average for June, July and August of each year, not the whole year.
Line 199 – Correlation is not equivalent to causation. Be careful with word choice: “confirming” is to strong.
Need to discuss: Why would a one month lag between dust arrival and Sargassum concentration be appropriate? That Sargassum is moving. Does a one-month lag make sense with the speed of the Caribbean current and transport of Sargassum through the region?
Lines 209-212 – Confusing sentence. Is the implication that increased chla, which is a proxy for phytoplankton concentration, can support Sargassum growth? Both are primary producers. There is also a citation missing.
There is no discussion of the Results and the Conclusions section is just a restatement of the Results. The Conclusions are repetitious of the Results and don’t add anything to the manuscript.
Then, there is a statement about the importance of monitoring dust and chla to help predict Sargassum. This case might be made for dust, since a one-month lag was suggested, but there is no reason why monitoring immediate chla concentrations could be useful in predicting Sargassum, when the Sargassum is likely already there.
Author Response
Please find responses to the reviewer's comments in the attached document.

Reviewer 2 Report
Comments and Suggestions for Authors
Line 3, Line 33, "Sargassum" should be in italics.
Line 34, should be amended to include a necessary keyword.
Line 48, in the title of the figure, it is also necessary to indicate the data source from which the figure was obtained (add necessary references).
This manuscript has a limited number of references. Some content related to the golden tide can also be referenced from research papers of other countries; for example, lines 71-74, I recommend 'DOI: 10.1016/j.scitotenv.2021.145726', 'DOI: 10.3390/jmse11030479', 'DOI: 10.1016/j.hal.2023.102451', 'DOI: 10.3390/jmse11010009'. Similarly, other references related to Sargassum can be used elsewhere.
Line 85, the research objective of this study needs to be briefly mentioned..
Line 86, please confirm whether 'Data & Methods' is a commonly used subtitle for this journal.
Line 86-line 135, this section needs to be divided into several subheadings.
Line 137-line 223, the subheadings need to be numbered. At the same time, this section should belong to the "Results and Discussion" section, and your main heading needs to be improved.
Line 224-256 of the Conclusion section do not require as many paragraphs. You can consolidate or combine them.
Please note that the genus name "Sargassum" should be used in italics. All species and genus names should be in italics. The entire document pays very little attention to the issue of italic formatting.
Line 293, line 295, line 323, are missing Article Number or page range. Please check the rest of the document as well.
Author Response
Please find the responses to the reviewer's comments in the attached document.

Round 2
Reviewer 1 Report
Comments and Suggestions for Authors
The manuscript is improved, but still needs minor revisions to work on qualifying some statements in the Results.
Abstract -
Line 24: Qualify the statement by adding “potential” before “contribution of elevated. . . .”
Lines 26-27: First mention of CHLO. Need to include it in the methods description portion of the abstract as well.
Introduction –
Lines 57-62: What do you mean by “both”? The paragraph state “on the one hand”, but then does not follow up with a counterpoint. This information in this paragraph seems to be now better delivered in the new paragraph below (Lines 71-79). I don’t think both paragraphs are necessary at this point. Also, why does Sargassum start with a lower case “s” in this paragraph?
Line 81: This is misleading. Chlorophyll is not a “driver” of Sargassum blooms.
Methods –
Line 130: Swap out “agglomerations” for “concentrations”. Doesn’t matter if the Sargassum is aggregated, just if it is in greater abundance.
Results and Discussion –
Lines 215-216: Still strong. Need to add “likely” or something like that to “enhanced some of the recent Sargassum outbreaks”.
Lines 223-224: Again, need to qualify this statement. “DMCD is one of the most important predictors of SCT” that we examined! The authors only
Line 227: Still incorrect to state that chlorophyll “can indirectly support the growth of Sargassum”.
Lines 228-230: Need to capitalize Sargassum.
Lines 236-237: Need to qualify this statement to acknowledge that they are the most important of the factors that were examined.
Line 265: Instead of “emphasize the role”, I suggest “support that hypothesis that Saharan dust contributes to Sargassum outbreaks”.
The additional text about limitations and future directions is necessary and helpful.
Author Response
Find responses to the reviewer in the attached document.
